# Haplotype and Haplotype-Environment Interaction Analysis Revealed Roles of SPRY2 for NSCL/P among Chinese Populations

**DOI:** 10.3390/ijerph16040557

**Published:** 2019-02-15

**Authors:** Ren Zhou, Mengying Wang, Wenyong Li, Siyue Wang, Hongchen Zheng, Zhibo Zhou, Yonghua Hu, Jing Li, Tao Wu, Hongping Zhu, Terri H. Beaty

**Affiliations:** 1School of Public Health, Peking University, Beijing 100191, China; zhouren1994@163.com (R.Z.); mywang@bjmu.edu.cn (M.W.); lwy_edu@163.com (W.L.); yzmagiw@163.com (S.W.); zhc_1995@163.com (H.Z.); yhhu@bjmu.edu.cn (Y.H.); jli@bjmu.edu.cn (J.L.); 2School of Stomatology, Peking University, Beijing 100081, China; zzbooo@126.com; 3School of Public Health, Johns Hopkins University, Baltimore, MD 21218, USA; tbeaty1@jhu.edu

**Keywords:** haplotypes, genetic association study, gene-environmental interaction, cleft lip and palate, case-parent trios

## Abstract

Non-syndromic cleft lip with or without cleft palate (NSCL/P) is one of common birth defects in China, with genetic and environmental components contributing to the etiology. Genome wide association studies (GWASs) have identified *SPRY1* and *SPRY2* to be associated with NSCL/P among Chinese populations. This study aimed to further explore potential genetic effect and gene—environment interaction among *SPRY* genes based on haplotype analysis, using 806 Chinese case—parent NSCL/P trios drawn from an international consortium which conducted a genome-wide association study. After the process of quality control, 190 single nucleotide polymorphisms (SNPs) of *SPRY* genes were included for analyses. Haplotype and haplotype—environment interaction analyses were conducted in Population-Based Association Test (PBAT) software. A 2-SNP haplotype and three 3-SNP haplotypes showed a significant association with the risk of NSCL/P after Bonferroni correction (corrected significance level = 2.6 × 10^−4^). Moreover, haplotype—environment interaction analysis identified these haplotypes respectively showing statistically significant interactions with maternal multivitamin supplementation or maternal environmental tobacco smoke. This study showed SPRY2 to be associated with NSCL/P among the Chinese population through not only gene effects, but also a gene—environment interaction, highlighting the importance of considering environmental exposures in the genetic etiological study of NSCL/P.

## 1. Introduction

Non-syndromic oral cleft (NSOC) is a complex disease with genetic and environmental factors together contributing to the etiology. The prevalence of NSOC varies across geographical regions or ethnic groups. Compared with Europeans and Africans, Chinese have a higher prevalence of 1.39~1.46/1000 live births, which is lower than the prevalence of American Indians and Japanese [1,2]. Genome wide association studies (GWASs) have contributed extraordinarily to gene mapping in the etiological exploration of NSCL/P, by identifying vast numbers of single nucleotide polymorphisms (SNPs) that are associated with diseases of interest. Despite the vast number of associations, only a small proportion of heritability could be explained by those identified variants from GWASs for many complex diseases [3]. It has been raised that individual SNP analysis might miss some genetic information due to the incomplete linkage disequilibrium (LD) between the tagged markers and causal variants, which then probably leads to a loss of efficiency in gene mapping [4,5]. Recently, evidence has increasingly demonstrated that a large proportion of genetic variance for complex diseases or traits could be explained by the genome-wide SNPs simultaneously [5,6], which implies that heritability is not actually missing, but instead undiscovered in GWAS by individual SNP analysis. Thus, further exploration of GWAS data were demanded to detect those remaining genetic risks.

Haplotype analysis generally contains more LD information compared with the individual SNP analysis [7]. This method has been applied to explore evidence for gene mapping using GWAS data, and successfully explained greater genetic variance than individual SNP analysis [8,9,10]. Moreover, haplotype analysis can also explore *cis*—interaction in the same quantitative trait loci (QTL) [11], which may facilitate the understanding about the genetic architecture of diseases. It is therefore reasonable to use haplotype analysis to excavate the remaining heritability in GWAS data. 

*SPRY* genes were recently identified as playing roles in the etiology of NSCL/P. Previous GWASs reported rs908822 (*SPRY1*), rs8001641 (*SPRY2*), and rs9545308 (*SPRY2*) showing a significant association among different populations, including Chinese people [12,13]. Evidence of a gene—gene interaction was also shown in this gene family to be associated with NSCL/P. In a recent published study, multiple pairs of SNP—SNP interactions among *SPRY1-SPRY2* and *SPRY2-SPRY4* contributed to the risk of NSCL/P in European and Asian populations, indicating the important roles of this gene family in the etiology of NSCL/P [14]. However, whether environment exposures participated in the relationship between *SPRY*s and NSCL/P has not been reported. Gene—environment interactions among *SPRY* genes and maternal environmental exposures should be explored to further clarify the roles of *SPRY* genes in the etiology of NSCL/P.

This study aimed to explore the potential genetic effect and gene—environment interaction among *SPRY* genes based on haplotype analysis, using GWAS data of 806 Chinese case-parent NSCL/P trios, which were recruited by an international consortium.

## 2. Materials and Methods 

### 2.1. Study Population

A total of 806 Chinese NSCL/P case-parent trios, recruited from Taiwan, Shandong, Hubei, and Sichuan, were drawn from an international consortium which conducted a genome-wide association study [15]. Of which, 539 cases were males and 267 were females; 597 cases had cleft lip and palate, and 209 cases had cleft lip only. Syndromic forms of CL/P were excluded by clinicians through clinical examination. The research protocol was reviewed and approved by the Institutional Review Boards (IRBs) of each institution participating in this international consortium, including IRBs at the Johns Hopkins School of Public Health, University of Pittsburgh, Utah State University, University of Iowa, Chang Gung Memorial Hospital, Wuhan University, Peking University, and West China School of Stomatology. The latest annual approval number was 91-06-10-03-2 approved by Johns Hopkins Bloomberg School of Public Health Committee on Human Research. Written informed consent was obtained from each participant and the study was performed in accordance with the Declaration of Helsinki. The whole blood, saliva, or mouthwash sample for cases and their parents were collected to extract DNA samples. 

### 2.2. Genotyping and SNP Selection

The process of DNA genotyping was conducted by the Center for Inherited Disease Research (CIDR) in the Johns Hopkins University, using Illumina Human610-Quad v.1_B Bead Chip. A total of 245 SNPs were genotyped in *SPRY* genes. The following four criteria were used to perform quality control: (1) Missing genotype information in the 806 trios >10%, (2) low minor allele frequency (MAF) among founders <0.05, (3) *p*-value of Hardy-Weinberg equilibrium test <0.001, (4) Mendelian errors >5%. After quality control, 55 SNPs were filtered due to a low minor allele frequency, leaving 190 SNPs for further analysis (shown in Appendix A). The process of quality control was conducted by PLINK (v1.07, http://pngu.mgh.harvard.edu/purcell/plink/) [16]. The hg38 version of the human genome was used to indicate the genomic coordinates.

### 2.3. The Assessment of Environmental Exposures

The information about maternal exposures was obtained through a face to face interview for each trio, including environmental tobacco smoke (ETS), maternal cigarette smoking, maternal alcohol consumption, and multivitamin supplementation from three months prior to pregnancy through the first trimester of pregnancy. The exposure status was recorded as 0 or 1, corresponding to unexposed or exposed. As less than 3% of mothers in the study population were exposed to maternal cigarette smoking and alcohol consumption, only ETS and multivitamin supplementation were included in haplotype—environment interaction analysis, with exposure rates of 40.9% and 12.9%, respectively.

### 2.4. Statistical Methods

Haplotype and haplotype—environment (HxE) interaction analysis were both performed using PBAT (population-based association test) software (v3.60, Harvard University, Boston, MA, USA) [17]. The statistical basis of PBAT was an approach introduced by Rabinowitz and Laird et al. [18,19], in which a score test was used to compare the observed genotypes among offspring with the expected distribution under Mendel’s law of segregation and the null hypothesis of ‘no linkage and no association’ [18,19]. PBAT could also incorporate the gene/covariate interactions into the test to further investigate the haplotype—environment interaction. Analyses were conducted under an additive genetic model. The method of sliding windows was used to determine the haplotypes for analyses, including 2-SNP haplotypes and 3-SNP haplotypes. Haplotypes with a frequency lower than 5% were excluded when reporting the results. To minimize the false positive results, Bonferroni correction was applied and the significance level was set as 2.6 × 10^−4^ (0.05/190). 

## 3. Results

A total of 806 NSCL/P trios were included in the haplotype analysis and haplotype—environment interaction analysis. 

The haplotype analysis under the additive genetic model yielded one 2-SNP haplotype and three 3-SNP haplotypes showing a significant association with NSCL/P among 806 Chinese trios after Bonferroni correction. The haplotype with the lowest *p*-value was T-A-G for rs541731-rs9531050-rs9545409 (*p*-value = 1.30 × 10^−5^), located in *SPRY2*, with a frequency of 5.7% (Table 1).

Haplotype—environment interaction analysis tested for the potential interaction between haplotypes in *SPRY* genes and maternal multivitamin supplementation as well as maternal environmental tobacco smoke (ETS). The analysis yielded a 3-SNP haplotype showing statistical interactions with maternal multivitamin supplementation: A-A-A for rs496932-rs9545412-rs9669948 (*p*-value = 3.43 × 10^−5^), located in *SPRY2*, with a frequency of 10.9%. For the interaction with maternal ETS, a 2-SNP haplotype and two 3-SNP haplotypes showed significant interactions with ETS after Bonferroni correction, with T-A-G for rs541731-rs9531050-rs9545409 showing the lowest *p*-value (*p*-value = 9.18 × 10^−5^). Table 2 presents the results of the interaction analysis.

## 4. Discussion

*SPRY* genes have been suggested in previous studies to participate in the development of NSCL/P. In this study, haplotype analysis and haplotype-environment interaction analyses were conducted to further explore the evidence of the *SPRY* genes in the etiology of NSCL/P, using 806 Chinese trios. A 2-SNP haplotype and three 3-SNP haplotypes showed a significant association with the risk of NSCL/P after Bonferroni correction. For the haplotype—environment interaction, these haplotypes respectively yielded statistically significant interactions with maternal multivitamin supplementation or maternal environmental tobacco smoke.

Several studies have previously shown evidence regarding the association between *SPRY* genes and NSCL/P. Ludwig et al. [13] first reported the *SPRY2* in 13q31 among a European population in a meta-analysis of GWASs, while the *p*-value for rs8001641 reached 6.51 × 10^−11^ and the homozygote relative risk was 2.41 (95%CI: 1.84–3.16). Then, Jia et al. [20] replicated the finding of rs8001641 using combined genotyping data of case-control samples and trios among a European population, but they failed to replicate the findings in Asian populations. Moreover, Yu et al. [12] found that rs908822 in *SPRY1* and rs9545308 in *SPRY2* showed significant associations with NSCL/P recently, using 3379 cases and 8593 controls in a Chinese Han population. Although Yu et al. did not replicate the first reported rs3005666 and rs8001641, respectively, in *SPRY1* and *SPRY2*, the novel markers identified in their study enriched the evidence of *SPRY* genes among the Chinese population, which implied that *SPRY* genes might influence the risk of NSCL/P not only among Europeans, but also among Chinese people or Asians. A recent study focused on gene—gene interactions among *SPRY* genes demonstrated that these genes contributed to the risk of NSCL/P through mutual interactions. Based on previous evidence of *SPRY* genes, the current study moved forward to explore potential susceptibility regions in *SPRY*s using haplotype analysis, as well as gene—environment interactions responsible for the risk of NSCL/P.

Based on data used in the current study, no individual SNP in *SPRY* genes showed a significant association with the risk of NSCL/P in the individual SNP analysis as well as the analysis of the SNP—environment interaction (data not shown). Target sequencing on SPRY genes could be a considerable approach to further explore potential variants associated with NSCL/P in these regions. Considering the higher statistical power of haplotype analysis compared to individual SNP analysis when LD was exploited to explore genes associated with diseases, we conducted the haplotype analysis and haplotype—environment interaction analysis to further investigate the roles of *SPRY* genes in Chinese NSCL/P. The results of haplotype analysis altogether indicated that one genetic segment (rs541731- rs9669948, from 80345577 to 80372188) located at 4.5 kb—31.1 kb downstream of *SPRY2* was associated with NSCL/P, indicating that the neighborhood region of this segment might contain causal variants. Although the position drifted a little compared with formerly reported positions (rs8001641, 222.4 kb upstream of *SPRY2*; rs9545308, 270.7 kb upstream of *SPRY2*), our findings still enriched the evidence of this gene in the etiology of NSCL/P. The biological function of *SPRY2* has also been identified by several experimental studies. Abnormal palates were observed in *SPRY2* knock-out mice [21,22,23], and the possible mechanism was suggested to be the regulation of the expression of the fibroblast growth factors (FGF) signaling pathway, which was shown to be important for the control of craniofacial development [12,24].

Gene—environment interactions have been suggested in several studies to influence the risk of NSCL/P, such as *IRF6*—multivitamin supplementation [25], and *RUNX2*—environmental tobacco smoke [26]. However, in most of the studies exploring potential SNP—environment factors’ interactions, the haplotype—environment interaction was rarely investigated. Multivitamin supplementation was previously identified as a protective factor of non-syndromic oral clefts, whereas environmental tobacco smoke was shown to be a risk factor [27,28]. In the current study, the haplotype—environment interaction analysis identified several haplotypes in *SPRY2* showing the statistical interactions with maternal multivitamin supplementation or ETS. Besides, these haplotypes were those that were significant in the haplotype analysis, which eliminated the probability that our results were accidentally significant. However, the similarity did not mean the roles of environmental factors could be ignored. On the contrary, this similarity and the interaction signals implied that the effect size of the association between these haplotypes and the risk of NSCL/P might be changed when environmental exposures were present. When exposed to harmful environmental factors, ones carrying the haplotype could suffer a higher risk of NSCL/P compared to those unexposed to the environmental factors. In such a situation, the similarity and interaction signals could provide evidence for practical preventions against environmental factors. Although statistical interactions do not always guarantee biological interaction, evidence about the biological function of *SPRY2* [21,29,30] and the signals in the single haplotype analysis might together support a biological interaction. Moreover, four SNPs in the identified genetic region, respectively, rs541731, rs496932, rs9545412, and rs9669948, are enhancer histone marks in embryonic stem cells or embryonic stem derived cells, which play important roles in the development of NSCL/P, as reported by ChromHMM software [31]. Genes with histone marks have been reported to have a significantly higher expression, while gene expression levels are correlated with DNA methylation [32]. Based on the general understanding that DNA methylations can be influenced by environmental exposures, it was reasonable to infer that ETS or multivitamin supplementation affected the DNA methylation processes of *SPRY2* in embryonic stem cells or embryonic stem derived cells, and then influenced the development of NSCL/P.

The case-parent design used in this study was robust to population stratification, which was an advantage compared to a case-control design. Besides, haplotypes contained more LD information and the haplotype analysis used in the current study could provide a higher statistical power for gene mapping, especially when the individual markers were not exactly the causal variants [7]. Despite these advantages, this study could not provide functional explanations for what we found, especially regarding how the haplotypes interacted with multivitamin supplementation and ETS. Thus, further studies are required to validate the findings as well as reveal the mechanism of how *SPRY2* influences the risk of NSCL/P, both individually and under certain environmental exposures.

## 5. Conclusions

This study identified that *SPRY2* was associated with the risk of NSCL/P among the Chinese population, and could interact with multivitamin supplementation and environmental tobacco smoke. It highlighted the importance of using haplotype analysis to explore the effect of genes and gene—environment interactions when no signal can be found in individual SNP analyses. Our study enriches the understanding of the role of *SPRY2* in the development of NSCL/P, and underlines the importance of considering environmental exposures in the etiological study of NSCL/P.

## Figures and Tables

**Table 1 ijerph-16-00557-t001:** The haplotypes in *SPRY* genes showing significant association with the risk of NSCL/P among 806 Chinese trios after Bonferroni correction ^1^.

Haplotypes	SNP Combination	Haplotype	Gene	Frequency (%)	*p*-value
2-SNP haplotype	rs9531050-rs9545409	A-G	*SPRY2*	7.3	1.75 × 10^−5^
3-SNP haplotype	rs541731-rs9531050-rs9545409	T-A-G	*SPRY2*	5.7	1.30 × 10^−5^
	rs9531050-rs9545409-rs496932	A-G-A	*SPRY2*	5.6	3.53 × 10^−5^
	rs496932-rs9545412-rs9669948	A-A-A	*SPRY2*	10.9	2.55 × 10^−4^

^1^*p*-value threshold = 2.6 × 10^−4^.

**Table 2 ijerph-16-00557-t002:** The haplotypes in *SPRY* genes showing significant interactions with environment exposures among 806 Chinese NSCL/P trios after Bonferroni correction ^1^.

Environmental Exposures	SNP Combination	Haplotype	Gene	*p*-value
Multivitamin supplementation	rs496932-rs9545412-rs9669948	A-A-A	*SPRY2*	3.43 × 10^−5^
ETS	rs9531050-rs9545409	A-G	*SPRY2*	9.44 × 10^−5^
	rs541731-rs9531050-rs9545409	T-A-G	*SPRY2*	9.18 × 10^−5^
	rs9531050-rs9545409-rs496932	A-G-A	*SPRY2*	2.41 × 10^−4^

^1^*p*-value threshold = 2.6 × 10^−4^.

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
