# Peer review of "Haplotype and Haplotype-Environment Interaction Analysis Revealed Roles of SPRY2 for NSCL/P among Chinese Populations"

_ijerph, 2019, doi:10.3390/ijerph16040557_

Round 1
Reviewer 1 Report
There is an interesting paper describing genetic and environmental effects on the development of non-syndromic cleft lip with or without cleft palate (NSCL/P)
Some editorial work and involvement of a native speaker is suggested to improve the text.
A significant effect of interaction between specific genetic haplotypes and “maternal multivitamin intake” and “maternal exposure to environmental tobacco smoke” on the NSCL/P risk was found. However, under Methods no information on the assessment of those environmental factor was presented.
Author Response
Dear reviewer,
Thanks for the suggestions and comments on our manuscript. These comments will greatly enhance the quality of this manuscript. We carefully considered the issues you reminded us, and also polished our English according to your suggestion. Our responses to your comments were listed in this letter point by point. The issues were also addressed in the revised manuscript.
We deeply hope our responses are satisfactory. Thank you again for your consideration and we shall await a favorable response to the revision.
Sincerely,
Ren Zhou
Point 1: Some editorial work and involvement of a native speaker is suggested to improve the text.
Response 1: Thank you for pointing out the impropriety of our English. We have carefully checked and polished our English according to your suggestions. Relevant sentences were revised directly in the manuscript (highlighted yellow).
Point 2: Under Methods no information on the assessment of those environmental factor was presented.
Response 2: Thanks for your comment. We have added a section under Method to explain details about the assessment of environmental factors. The relevant sentences read as below:
‘The assessment of environmental exposures
The information about maternal exposures was obtained through face to face interview for each trio, including environmental tobacco smoke (ETS), maternal cigarette smoking, maternal alcohol consumption, and multivitamin supplementation from three months prior to pregnancy through the first trimester of pregnancy. The exposure status was recorded as 0 or 1, corresponding to unexposed or exposed. As merely less than 3% of mothers in the study population were exposed to maternal cigarette smoking and alcohol consumption, only ETS and multivitamin supplementation were included in haplotype-environment interaction analysis, with exposure rates of 40.9% and 12.9% respectively.’
This manuscript describes ostensible haplotype & haplotype-environment interaction analysis to identify the causative loci for cleft lip and/or palate. Generally speaking choosing two SNPs can not be called haplotype because the number of two markers is not enough for this genetic analysis. Also there are some concerns regarding how the data is presented. For instance, there is no Human Genome Browser version used to indicate the genomic coordinates.
In the Abstract, the two genomic coordinates (79817713 and 79844324) are not located at 4.6 kb-31.3 kb from 3' end of SPRY2. If these coordinates are based on hg19 version of Human Genome Browser, the distance would be 492 kb-518 kb. If these are based on hg38 version, the distance will be more than this. Within this interval, there are a couple of long non-coding RNAs and there is no reasonable explanation why one of these non-coding RNAs can not be a candidate gene for NSCL/P. In the Discussion, the authors say that one genetic segment (rs541731-rs9669948, from 79817713 to 79844324) might contain causal variant, which is unreasonable, because the causative variant is also likely located in the neighborhood of SNPs used for GWAS. The same is true for the statement that four SNPs are enhancer histone marks.
Especially this hypothesis is not acceptable, because only limited number of 190 SNPs near SPRY genes were used for this analysis. This study is biased, therefore, at the inception.
It was evidenced by the discrepancy of the candidate loci published before.
Also trying to find a locus is self-contradictory with the statement that NSCL/P is a complex disease, even if the environmental factors are considered.
Collectively, this data does not show convincing evidence of SPRY2 gene to be involved in NSCL/P.
Author Response
Dear reviewer,
Thanks for the suggestions and comments on our manuscript. These comments will greatly enhance the quality of this manuscript. We carefully considered the issues you reminded us, and then made some revisions accordingly. Our responses to your comments and questions were listed in this letter point by point, which were also addressed in the revised manuscript. We also polished our English according to your suggestion. Inappropriate English expressions were revised directly in the manuscript.
We deeply hope our responses are satisfactory. Thank you again for your consideration and we shall await a favorable response to the revision.
Sincerely,
Ren Zhou
Point 1: Generally speaking choosing two SNPs can not be called haplotype because the number of two markers is not enough for this genetic analysis.
Response 1: Thank you for this comment. Haplotypes including 2-4 SNPs were generally used for haplotype analysis in the etiological exploration of non-syndromic oral clefts[1-3]. Besides, it was also reported that size of 2-4 SNPs can basically meet the requirement of haplotype analysis[4]. To improve our analysis, we have conducted 5-SNP haplotype analysis. The results also indicated the same region as two-SNP or three-SNP haplotype analysis showed. The results of five-SNP haplotype analysis (shown below) were attached to the manuscript in the supplemental file.
Table S2. The 5-SNP haplotypes in SPRY genes showing significant association with the risk of NSCL/P or among 806 Chinese trios after Bonferroni correction1
SNP combination | Haplotype | Gene | Frequency (%) | p-value |
rs4728-rs504122-rs541731-rs9531050-rs9545409 | A-A-T-A-G | SPRY2 | 3.0 | 1.69×10-5 |
rs504122-rs541731-rs9531050-rs9545409-rs496932 | A-T-A-G-A | SPRY2 | 2.5 | 2.92×10-5 |
rs541731-rs9531050-rs9545409-rs496932-rs9545412 | T-A-G-A-A | SPRY2 | 4.8 | 3.96×10-5 |
rs1772587-rs9574697-rs10507911-rs1772573-rs1772569 | C-A-T-A-C | SPRY2 | 0.4 | 4.81×10-5 |
rs9545412-rs9669948-rs486467-rs502700-rs534870 | A-G-A-T-G | SPRY2 | 4.2 | 1.21×10-4 |
rs9531050-rs9545409-rs496932-rs9545412-rs9669948 | A-G-A-A-G | SPRY2 | 4.4 | 1.38×10-4 |
rs954771-rs7989054-rs1772587-rs9574697-rs10507911 | C-G-C-A-T: | SPRY2 | 0.2 | 1.67×10-4 |
rs7989054-rs1772587-rs9574697-rs10507911-rs1772573 | G-C-A-T-A | SPRY2 | 0.2 | 1.75×10-4 |
1 p-value threshold= 2.6×10-4.
References:
1. Mostowska, A.; Hozyasz, K. K.; Biedziak, B.; Wojcicki, P.; Lianeri, M.; Jagodzinski, P. P., Genotype and haplotype analysis of WNT genes in non-syndromic cleft lip with or without cleft palate. Eur J Oral Sci 2012, 120 (1), 1-8.
2. Wang, Y.; Sun, Y.; Huang, Y.; Pan, Y.; Shi, B.; Ma, J.; Ma, L.; Lan, F.; Zhou, Y.; Shi, J.; Zhu, J.; Jiang, H.; Zhang, L.; Xiao, X.; Jiang, M.; Yin, A.; Yu, L.; Wang, L.; Cheng, J.; Yang, Y., The association study of nonsyndromic cleft lip with or without cleft palate identified risk variants of the GLI3 gene in a Chinese population. J Genet 2017, 96 (4), 687-693.
3. Skare, O.; Gjessing, H. K.; Gjerdevik, M.; Haaland, O. A.; Romanowska, J.; Lie, R. T.; Jugessur, A., A new approach to chromosome-wide analysis of X-linked markers identifies new associations in Asian and European case-parent triads of orofacial clefts. PloS one 2017, 12 (9), e0183772.
4. Khankhanian, P.; Gourraud, P. A.; Lizee, A.; Goodin, D. S., Haplotype-based approach to known MS-associated regions increases the amount of explained risk. J Med Genet 2015, 52 (9), 587-94.
Point 2: There is no Human Genome Browser version used to indicate the genomic coordinates.
Response 2: Thank you for pointing out this problem. We added the Human Genome Browser version of the GWAS data in ‘Method’. The corresponding sentence now reads:
‘The Ensemble release 54 (http://may2009.archive.ensembl.org/Homo_sapiens/ Info/Index), based on the National Center for Biotechnology Information Human Genome build 36 which was applied in this study, was used to indicate the genomic coordinates.’
Point 3: In the Abstract, the two genomic coordinates (79817713 and 79844324) are not located at 4.6 kb-31.3 kb from 3' end of SPRY2. If these coordinates are based on hg19 version of Human Genome Browser, the distance would be 492 kb-518 kb. If these are based on hg38 version, the distance will be more than this. Within this interval, there are a couple of long non-coding RNAs and there is no reasonable explanation why one of these non-coding RNAs can not be a candidate gene for NSCL/P.
Response 3: Thank you for this comment. Since the original GWAS study which our data derived from mapped SNPs according to National Center for Biotechnology Information Human Genome build 36.1, we then used Ensemble release 54 (based on NCBI 36) to map the genomic coordinates, indicating 13:79817713-79844324 were located at 4.6kb-31.3kb from 3’ end of SPRY2.
Point 4: In the Discussion, the authors say that one genetic segment (rs541731-rs9669948, from 79817713 to 79844324) might contain causal variant, which is unreasonable, because the causative variant is also likely located in the neighborhood of SNPs used for GWAS. The same is true for the statement that four SNPs are enhancer histone marks.
Response 4: Thank you for this comment. We agree with you that some expressions in the manuscript were inappropriate. We have carefully revised relevant sentences and they now read:
‘The results of haplotype analysis altogether indicated that one genetic segment (rs541731- rs9669948, from 79817713 to 79844324) located at 4.6kb-31.2kb 3’ of SPRY2 was associated with NSCL/P, indicating that the genetic segment together with the neighborhood region might contain causal variants. Although the position drifted a little compared with formerly reported positions (rs8001641, 217 kb 3' of SPRY2; rs9545308, 217 kb 3' of SPRY2), our findings still enriched the evidence of this gene in etiology of NSCL/P.’
Point 5: The hypothesis is not acceptable, because only limited number of 190 SNPs near SPRY genes were used for this analysis
Response 5: Thank you for this comment. Since the genotyping data in this study was derived from a GWAS, only 245 SNPs were genotyped in the original study and the quality control process left 190 SNPs for analysis. We have tried to impute data for non-genotyping alleles, but the imputed data only supplement twelve SNPs which located away from the association signals in our study. Moreover, in terms of haplotype analysis applied in this study, this method could capture more information about the neighborhood non-genotyping SNPs to increased power. Therefore, we decided not to expand the analysis to include the imputed data.
Point 6: Trying to find a locus is self-contradictory with the statement that NSCL/P is a complex disease, even if the environmental factors are considered
Response 6: Thank you for this comment. We agree with you that NSCL/P is a complex disease with genetic and environmental factors contributing to the etiology. Many genes and environmental factors could participate in the development of disease. Nevertheless, the aim of this study was to explore potential genes which might explain a proportion of the etiology, rather than to find the only one causal gene.
Reviewer 3 Report
1) Current publications use terminology SNV instead of SNP. Consider revising throughout 2) Introduction: birth prevalence data is unclear; there are other ethnic groups with a higher prevalence than China. 3) Methods: Percentage of mothers with self-reported alcohol use is below normal published data (in a general population). This is not later discussed. What percentage of Chinese mothers (in general population) report alcohol use during pregnancy? Is 3% less than that? Is the multi-vitamin and ETS exposure typical in this population. 12.9% vitamin usage seems lower than expected. 4) Methods: What other institutions were included (don't list some and not others) 5) Results: If 5% frequency of haplotypes is your cut-off, when you stratify your environmental exposures, what frequency of positive exposure contain the haplotype of interest? I'm concerned that you may be overstating your results based on a small percentage of your population. 6) Discussion: First paragraph, 3rd sentence: three 2-SNP and six 3-SNP haplotypes - I think this is incorrect - Table 1 and 2 only have two 2-SNV results. Next sentence states environmental exposure haplotypes. Over representation of your data. Either clearly state that "overall" and "of these, # were from the environmental exposure results". Two SNV results are contrary - overall AG and ETS is AC. Reasons/explanations should be discussed. Difference may highlight an environmental affect. Three SNV results have similarities (AAA in both analyses for one, AGA in both for another, TAG in both for another). Similarity does not reflect a difference based on ethnicity. Environmental affect, while important, appears overstated in discussion as it currently is written. 7) LD plots should be included (even as supplemental) for all 4 genes. 8) Concerned that no single SNV was associated in this population. Has targeted sequencing been considered in the regions recommended in discussion?This could add to this paper significantly.
Author Response
Dear reviewer,
Thanks for the suggestions and comments on our manuscript. These comments will greatly enhance the quality of this manuscript. We carefully considered the issues you reminded us, and then made some revisions accordingly. Our responses to your comments and questions were listed in this letter point by point, which were also addressed in the revised manuscript. We also polished our English according to your suggestion. Inappropriate English expressions were revised directly in the manuscript.
We deeply hope our responses are satisfactory. Thank you again for your consideration and we shall await a favorable response to the revision.
Sincerely,
Ren Zhou
Point 1: Current publications use terminology SNV instead of SNP.
Response 1: Thank you for the suggestion. Compared to SNPs, SNVs additionally contain variants with low frequency. Since our data is based on GWAS and common variants were included for analysis, we think ‘SNP’ might be better and decide not to replace it.
Point 2: Introduction: birth prevalence data is unclear; there are other ethnic groups with a higher prevalence than China.
Response 2: Thank you for the comment. We agree with you that the epidemiology of this disease in different ethnicities should be explained in details. Birth prevalence data for different ethnic groups have been added in the manuscript. Relevant sentences now read:
‘Non-syndromic oral cleft(NSOC) is a complex disease with genetic and environmental factors together contributing to the etiology. The prevalence of NSOC varies across geographical regions or ethnic groups. Compared with Europeans and Africans, Chinese had a higher prevalence of 1.39~1.46/1000 live births, which was lower than the prevalence of American Indians and Japanese [1-2].’
References:
1. Cooper, M. E.; Ratay, J. S.; Marazita, M. L., Asian oral-facial cleft birth prevalence. The Cleft palate-craniofacial journal: official publication of the American Cleft Palate-Craniofacial Association 2006, 43 (5), 580-9.
Panamonta, V.; Pradubwong, S.; Panamonta, M.; Chowchuen, B., Global Birth Prevalence of Orofacial Clefts: A Systematic Review. J Med Assoc Thai 2015, 98 Suppl 7, S11-21.
Point 3: Methods: Percentage of mothers with self-reported alcohol use is below normal published data (in a general population). This is not later discussed. What percentage of Chinese mothers (in general population) report alcohol use during pregnancy? Is 3% less than that? Is the multi-vitamin and ETS exposure typical in this population. 12.9% vitamin usage seems lower than expected.
Response 3: Thank you for this comment. The percentages of environmental exposure in this study were concordance with the level in general Chinese pregnant women. A hospital-based study which investigated lifestyle during pregnancy of new mothers at their hospital discharges reported that more than 97% has never drank alcohol throughout the pregnancy [3]. Besides, the China Kadoorie Biobank (CKB) study on half a million Chinese people reported that prevalence rates of regular drinking among women (not limited in pregnant women) was 2.2% [4]. Although the CKB data did not focus on pregnancy women, it indeed reflected that the alcohol exposure proportion for Chinese female was low. Based on the data shown above, the proportion of maternal alcohol use in our study was concordance with the level of general Chinese pregnancy women.
As for the multivitamin supplement, a study conducted in two cities reported nearly 20% of pregnant women used nutrients supplement [5]. Another study covering 28,000 women showed the percentage of iron-containing multi-micronutrients use was less than 20% in 2010-2013 [6]. Since the percentage of nutrients supplement was generally higher in cities than in rural areas, which increased through years, 12.9% of multivitamin supplement was reasonable among the sample.
References:
Tang, L.; Lee, A. H.; Binns, C. W.; Hui, Y. V.; Yau, K. K. W., Consumption of Chinese herbal medicines during pregnancy and postpartum: A prospective cohort study in China. Midwifery 2016, 34, 205-210.
Lyu, J.; Guo, Y.; Bian, Z.; Yu, C.; Wang, Z.; Zhou, H.; Tan, Y.; Chen, J.; Chen, Z.; Li, L.; China Kadoorie Biobank Collaborative, G.; China Kadoorie Biobank, C. K. B. C. G., [Regional differences in patterns of alcohol consumption: findings from the China Kadoorie Biobank study on half a million people from 10 regions]. Zhonghua Liu Xing Bing Xue Za Zhi 2014, 35 (8), 875-81.
Jing, J.; Xia, H.; Gao, J.; Jin, Y.; Ling, L., Comparision study on nutrients supplement among pregnant women. Zhongguo Shi Yong Hu Li Za Zhi 2010, 26 (4), 69-72.
Liu, D. M.; Li, J. M.; Qu, P. F.; Dang, S. N.; Wu, X. Y.; Zhang, R.; Yan, H.; Yan, H., [Prenatal supplementations of iron, iron-containing multimicronutrients and antianemic Chinese patent medicines in women in Shaanxi province, 2010-2013]. Zhonghua Liu Xing Bing Xue Za Zhi 2017, 38 (11), 1466-1470.
Point 4: Methods: What other institutions were included (don't list some and not others)
Response 4: Thank you for pointing out this problem. We have listed the other institutions in the manuscript. Now the sentence read:
‘The research protocol had been reviewed and approved by Institutional Review Boards (IRBs) of each institution participating in this international consortium, including IRBs at the Johns Hopkins School of Public Health, University of Pittsburgh, Utah State University, University of Iowa, Chang Gung Memorial Hospital, Wuhan University, Peking University and West China School of Stomatology.’
Point 5: Results: If 5% frequency of haplotypes is your cut-off, when you stratify your environmental exposures, what frequency of positive exposure contain the haplotype of interest? I'm concerned that you may be overstating your results based on a small percentage of your population.
Response 5: Thank you for this question. For multivitamin supplement, the proportion of carrying haplotypes of interest among positive exposure were 7.1%, 6.0%, 7.1% and 13.1%, respectively for rs9531050-rs9545409 (A-G), rs541731-rs9531050-rs9545409 (T-A-G), rs9531050-rs9545409-rs496932 (A-G-A) and rs496932-rs9545412-rs9669948 (A-A-A). For ETS, the proportion of carrying haplotypes of interest among positive exposure were 8.3%, 7.4%, 7.7% and 10.5%, respectively for rs9531050-rs9545409 (A-G), rs541731-rs9531050-rs9545409 (T-A-G), rs9531050-rs9545409-rs496932 (A-G-A) and rs496932-rs9545412-rs9669948 (A-A-A).
Point 6: Discussion: First paragraph, 3rd sentence: three 2-SNP and six 3-SNP haplotypes - I think this is incorrect - Table 1 and 2 only have two 2-SNV results. Next sentence states environmental exposure haplotypes. Over representation of your data. Either clearly state that "overall" and "of these, # were from the environmental exposure results". Two SNV results are contrary - overall AG and ETS is AC. Reasons/explanations should be discussed. Difference may highlight an environmental affect. Three SNV results have similarities (AAA in both analyses for one, AGA in both for another, TAG in both for another). Similarity does not reflect a difference based on ethnicity. Environmental affect, while important, appears overstated in discussion as it currently is written.
Response 6: Thank you for this comment and pointing out the incorrect numbers. We have revised the sentence and it now reads:
‘A 2-SNP haplotype and three 3-SNP haplotypes showed significant association with the risk of NSCL/P after Bonferroni correction. For haplotype-environment interaction, these haplotypes respectively yielded statistically significant interactions with maternal multivitamin supplementation or maternal environmental tobacco smoke.’
For the difference allelic information between haplotype analysis (A-G) and haplotype-environment interaction (A-C), we rechecked the results and found a mistake might occur when we displayed the results. The 2-SNP combination for haplotype-environment interaction was also A-G, the same as haplotype analysis.
As for the three SNVs showing similar results when environmental exposures were included or not, firstly, the similarities eliminated the probability that our results are accidentally significant. Secondly, the similarity did not mean the roles of environmental factors could be ignored. Although the haplotypes could have potential effect on NSCL/P in the absence of environmental exposures, the effect size of the association between these haplotypes and the risk of NSCL/P could be changed when environmental exposures were present. When exposed to harmful environmental factors, ones carrying the haplotype could suffer higher risk of NSCL/P compared to unexposed to the environmental factors. We explained this issue in detail in Discussion, and relevant sentences read:
‘In the current study, the haplotype-environment interaction analysis identified several haplotypes in SPRY2 showing the statistical interactions with maternal multivitamin supplementation or ETS. Besides, these haplotypes were those showing significant in haplotype analysis, which eliminated the probability that our results are accidentally significant. However, the similarity did not mean the roles of environmental factors could be ignored. On the contrary, this similarity and the interaction signals implied that the effect size of the association between these haplotypes and the risk of NSCL/P might be changed when environmental exposures were present. When exposed to harmful environmental factors, ones carrying the haplotype could suffer higher risk of NSCL/P compared to unexposed to the environmental factors. In such situation, the similarity and interaction signals could nicely provide evidence for practical preventions for environmental factors.’
Point 7: LD plots should be included (even as supplemental).
Response 7: Thank you for this comment. LD plots (shown below) have been added as supplemental files.
Figure S1. LD plot for single nucleotide variations (SNVs) in SPRY1
Figure S2. LD plot for single nucleotide variations (SNVs) in SPRY2
Figure S3. LD plot for single nucleotide variations (SNVs) in SPRY4
Point 8: Concerned that no single SNV was associated in this population. Has targeted sequencing been considered in the regions recommended in discussion? This could add to this paper significantly.
Response 8: Thanks for your question and suggestion. Targeted sequencing was a considerable approach to further explore variants associated with NSCL/P in this region. We have done some work and are analyzing the sequencing data. According to your suggestion, we added some discussion about it and the sentences read:
‘Based on data used in the current study, no individual SNP in SPRY genes showed significant association with the risk of NSCL/P in the individual SNP analysis as well as the analysis of SNP-environment interaction (data not shown). Target sequencing on SPRY genes could be a considerable approach to further explore potential variants associated with NSCL/P in these regions.’
Round 2
Reviewer 2 Report
The revised manuscript considerably enhanced the description of the study.
Nonetheless, this reviewer has the following concern in the Point 3.
The authors said they used NCBI36 (hg18) version of Human Genome Browser.
The problem is that there have been many corrections and improvements of human genome sequence since hg18 version, which was released in 2006. They have to use more updated version like hg38 which was released in 2013. Based on this improved version the genomic segment associated to NSCL/P, the associated region is much far away from the gene SPRY2.
The authors should use hg38 version for the genomic coordinates and describe the distance between the associated segment and SPRY2 correctly. Also the segment is not from the 3' end of SPRY2 gene. It is located distal from 5' end of SPRY2. This error should be corrected too.
Author Response
Point 1: The authors said they used NCBI36 (hg18) version of Human Genome Browser. The problem is that there have been many corrections and improvements of human genome sequence since hg18 version, which was released in 2006. They have to use more updated version like hg38 which was released in 2013. Based on this improved version the genomic segment associated to NSCL/P, the associated region is much far away from the gene SPRY2.The authors should use hg38 version for the genomic coordinates and describe the distance between the associated segment and SPRY2 correctly. Also the segment is not from the 3' end of SPRY2 gene. It is located distal from 5' end of SPRY2. This error should be corrected too.
Response 1: Thank you for the comment and suggestion. According to your suggestion, we revised the manuscript using the hg38 version for the genomic coordinates to present the positions of the SNPs.
To be more specific, SPRY gene was located at chr13:80335976-80341115, while the associated segment located from chr13:80345577 to chr13:80372188, 4.5kb-31.1kb downstream of SPRY2.
We revise relevant sentences in Method and the third paragraph of Discussion and they now read:
‘The hg38 version of human genome was used to indicate the genomic coordinates.’
‘The results of haplotype analysis altogether indicated that one genetic segment (rs541731- rs9669948, from 80345577 to 80372188) located at 4.5kb-31.1kb downstream of SPRY2 was associated with NSCL/P, indicating that the neighborhood region of this segment might contain causal variants. Although the position drifted a little compared with formerly reported positions (rs8001641, 222.4 kb upstream of SPRY2; rs9545308, 270.7 kb upstream of SPRY2), our findings still enriched the evidence of this gene in etiology of NSCL/P.’